# Pharmacological Preconditioning with Fenofibrate in Cardiomyocyte Cultures of Neonatal Rats Subjected to Hypoxia/Reoxygenation, High Glucose, and Their Combination

**DOI:** 10.3390/ijms252111391

**Published:** 2024-10-23

**Authors:** Víctor Hugo Oidor-Chan, Araceli Sánchez-López, Agustina Cano-Martinez, Willy Ramses García-Niño, Elizabeth Soria-Castro, Leonardo del Valle-Mondragón, Gabriela Zarco-Olvera, Mariana Patlán, Veronica Guarner-Lans, Emma Rodríguez-Maldonado, Javier Flores-Estrada, Vicente Castrejón-Téllez, Luz Ibarra-Lara

**Affiliations:** 1Department of Biotechnology, Autonomous Metropolitan University, Iztapalapa Campus, Av. Ferrocarril de San Rafael Atlixco 186, Col. Leyes de Reforma 1ª. Sección, Alcaldía Iztapalapa, Mexico City 09310, Mexico; victorhugooidor@xanum.uam.mx; 2Department of Pharmacobiology, Center for Research and Advanced Studies of the National Polytechnic Institute, Calz. de los Tenorios 235, Col Granjas Coapa, Tlalpan, Mexico City 14330, Mexico; asanchezl@cinvestav.mx; 3Department of Physiology, National Institute of Cardiology Ignacio Chávez, Juan Badiano No. 1, Col. Sección XVI, Tlalpan, Mexico City 14080, Mexico; agustina.cano@cardiologia.org.mx (A.C.-M.); veronica.guarner@cardiologia.org.mx (V.G.-L.); 4Department of Cardiovascular Biomedicine, National Institute of Cardiology Ignacio Chávez, Juan Badiano No. 1, Col. Sección XVI, Tlalpan, Mexico City 14080, Mexico; ramses.garcia@cardiologia.org.mx (W.R.G.-N.); elizabeth.soria@cardiologia.org.mx (E.S.-C.); 5Department of Pharmacology, National Institute of Cardiology Ignacio Chávez, Juan Badiano No. 1, Col. Sección XVI, Tlalpan, Mexico City 14080, Mexico; leonardo.delvalle@cardiologia.org.mx (L.d.V.-M.); gabriela.zarco@cardiologia.org.mx (G.Z.-O.); 6Subdirection of Basic and Technological Research, National Institute of Cardiology Ignacio Chávez, Juan Badiano No. 1, Col. Sección XVI, Tlalpan, Mexico City 14080, Mexico; mariana.patlan@cardiologia.org.mx; 7Laboratory of Cell Biology, Department of Physiology, National Institute of Cardiology Ignacio Chávez, Juan Badiano No. 1, Col. Sección XVI, Tlalpan, Mexico City 14080, Mexico; emma.rodriguez@cardiologia.org.mx; 8Division of Investigation, Juarez Hospital of Mexico, Av. Instituto Politecnico Nacional No. 5160, Magdalena de las Salinas, Gustavo A. Madero, Mexico City 07760, Mexico; jose.florese@salud.gob.mx

**Keywords:** pharmacological preconditioning, fenofibrate, hypoxia-reoxygenation, high glucose, apoptosis

## Abstract

Pharmacological preconditioning is an alternative to protect the heart against the consequences of damage from ischemia/reperfusion (I/R). It is based on the administration of specific drugs that imitate the effect of ischemic preconditioning (IPC). Peroxisomal proliferator-activated receptors (PPARs) can prevent apoptosis in pathologies such as I/R and heart failure. Therefore, our objective was to determine if the stimulation of PPARα with fenofibrate (feno) decreases the apoptotic process induced by hypoxia/reoxygenation (HR), high glucose (HG), and HR/HG. For that purpose, cardiomyocyte cultures were divided into the following groups: Group 1—control (Ctrl); Group 2—HR; Group 3—HR + 10 μM feno; Group 4—HG, (25 mM glucose); Group 5—HG + feno; Group 6—HR/HG, and Group 7—HR/HG + feno. Our results indicate that cell viability decreases in neonatal cardiomyocytes undergoing HR, HG, and their combination, while feno improved cell viability. Feno treatment decreased apoptosis compared with HG-, HR-, or HG/HR-vehicle-treated. Nuclear- and mitochondrial-apoptosis markers increased in neonatal cardiomyocytes from HR, HG, and HR/HG; while the cytotoxicity decreased in cells treated with feno. In addition, the expression of Bax, Bad, and caspase 9 decreased due to feno, while 14-3-3ɛ and Bcl2 were increased. Inner mitochondrial cytochrome C increased with feno in every condition, as well as mitochondrial activity. Feno treatment prevented injury in the ultrastructure and in the mitochondrial membranes. Thus, our results suggest that feno decreases apoptosis in neonatal cardiomyocytes, improving the ultrastructure of mitochondria in the pathological conditions studied.

## 1. Introduction

In 1986, Murry et al. described that short periods of coronary artery occlusion followed by reperfusion reduced cardiac infarct size after global ischemia. This phenomenon was termed ischemic preconditioning (IPC) [1]. However, due to its invasive nature, it is not practical to apply and, therefore, pharmacological preconditioning was proposed as another way to protect the heart against injury from the ischemia/reperfusion (I/R). The concept is based on the administration of specific drugs such as volatile anesthetics, noble gases, opioids, propofol, dexmedetomidine, and phosphodiesterase inhibitors which mimic the effect of IPC [2].

I/R injury consists of cell death generated by the restoration of the blood supply to the myocardium after an acute myocardial infarction (AMI). Oxidative stress, intracellular and mitochondrial calcium (Ca^2+^) overload, rapid restoration of physiological pH levels, and opening of the mitochondrial permeability transition pore (mPTP) are processes that contribute to I/R-induced cellular injury. They result in the release of cytochrome C (Cyt C) to form the apoptosome, which is a complex composed of procaspase-9 and its cofactor APAF-1, swelling of the mitochondrial matrix, release of proapoptotic factors (Bax, Bad, and Bak), and cell death [3,4].

Hyperglycemia increases the risk of death and promotes poor outcomes after AMI as observed in numerous ex-vivo, animal, and clinical studies, showing that hyperglycemia may mediate adverse effects on inflammation, cell injury, apoptosis, myocardial ischemic metabolism, endothelial function, the coagulation cascade, and increased platelet aggregation acute ischemia [5,6,7]. In studies conducted in our laboratory, we observed that pharmacological preconditioning with the peroxisomal proliferator-activated receptor-alpha (PPARα) agonist fenofibrate (feno) improves cell viability and favors an antioxidant environment that contributes to preserve cardiac ultrastructure under conditions of hyperglycemia, hypoxia/reoxygenation (H/R), or both [8]. We also observed that PPARα activation optimizes cardiac metabolism, improves vasodilation, and preserves cardiac structure during I/R in the heart of rats with metabolic syndrome [9]. Moreover, the PPARα agonist WY-14643, administered acutely to rats, generated cardioprotection after I/R ex vivo, and there was an improvement of the contractile dysfunction and a decrease in the generation of arrhythmias. This cardioprotection involved the participation of the phosphatidylinositol 3-kinase (PI3K)/protein kinase B (Akt) pathway [10]. In another study, Bulhak et al. reported that PPARα activation by acute administration of WY-14643 exerted cardioprotection in an AMI model generated in Goto-Kakizaki type 2 diabetic rats, involving mechanisms related to nitric oxide (NO) production through the PI3K/Akt pathway [11]. PPARα activation can also confer delayed cardioprotection similar to IPC in a rat I/R model by modulating oxidative stress, matrix metalloproteinase-2 activation, and Bcl-2 and Bax expression [12]. The evidence described indicates that activation of PPARα receptors could be a therapeutic alternative to generate cardioprotection from the damage generated by AMI. Therefore, the objective of our research was to evaluate whether pharmacological preconditioning with feno in primary cultures of neonatal rat heart cells subjected to HG, HR, or HG/HR, preserves cell viability, decreases cytotoxicity, and decreases the concentration and expression of apoptotic markers.

## 2. Results

### 2.1. Validation of HR Model

To explore if the method used to produce HR was effective, we evaluated the expression of HIF-1α. Our results show that the expression of HIF-1α increased in the cultured cells subjected to HR without undergoing 1 h of reoxygenation compared to control cells (Figure 1), suggesting that the method used was effective in producing hypoxia.

### 2.2. Cell Viability

Our results show that control cultured cells, independently of treatment (DMSO, fenofibrate, or mannitol) presented a 95% viability, indicating that they did not affect cell viability. Cells subjected to HR and HG had less cell viability compared to the control, suggesting that the mechanisms that were activated by the absence of oxygen damaged the cells. The combination of HR and HG is highly toxic for the cardiomyocytes. The feno treatment was able to prevent mortality of cells in HR, HG, and HR/HG groups, respectively. These results confirm that fenofibrate preserved cell viability despite HR or HG, but its effect was limited in cells exposed to HR/HG (Figure 2).

### 2.3. Fenofibrate Protects Cells from HR, HG, and Both Conditions (Determination of Cytotoxicity, 8-OH-2dG, and Malonate)

The hallmark process of cytotoxicity is the secretion of granules. They contain pore-forming proteins and apoptosis-related proteins. Our results show that cells subjected to HR, HG, or HR/HG treatments developed greater cytotoxicity, evidenced as a higher apoptosis rate and cell damage. Cells treated with feno exhibited lower cytotoxicity compared to cells exposed to an altered environment and vehicle treated (Figure 3A). To explore apoptosis-induced nuclear damage, we evaluated 8-hydroxy-2-deoxyguanosine (8-OH-2dG) (Figure 3B). Our results show that HR, HG, and HR/HG induced a raise in its expression, while the groups treated with feno exhibited a lower 8-OH-2dG value (Figure 3B). Malonate is a competitive inhibitor of succinic dehydrogenase, arresting the Krebs cycle. Succinic dehydrogenase catalyzes the reaction converting dehydrogenated succinate to fumarate. Malonate may induce apoptosis in mitochondria, since it damages mitochondrial cell respiration, inhibiting complex II, and preventing the oxidative phosphorylation reaction. Our results show that HR, HG, and HR/HG-subjected cell cultures exhibited an increased malonate production, while the groups treated with feno significantly decreased its production (Figure 3C). These results support the hypothesis that feno protects mitochondria and mitochondrial respiration.

### 2.4. Fenofibrate Decreases Apoptotic Cells in Pathological Conditions

Figure 4 shows representative images from cells exposed to the different conditions, where the green mark corresponds to cellular nuclei in apoptosis and the blue mark identifies viable nuclei. Under control conditions (Figure 4A), a certain degree of apoptosis was detected, while cultured cells with HR (Figure 4B), HG (Figure 4D), and the combination of both conditions (Figure 4F) presented a higher number of apoptotic cells. The treatment with feno reduced the number of apoptotic cells at all conditions (Figure 4C, Figure 4E, and Figure 4G, respectively). Therefore, feno treatment attenuates deleterious actions of HG and HR in primary neonatal cardiomyocytes cells culture.

### 2.5. Determination of Apoptosis Using Annexin V/PI Staining

We observed that the HR group shows slight levels of cellular necrosis and HR/HG groups present a greater apoptosis. Treatment with feno reversed necrosis and apoptosis and increased the number of live cells. Therefore, feno protects neonatal cardiomyocytes against apoptosis, necrosis, and cellular death (Figure 5).

### 2.6. Fenofibrate Promotes Survival Molecules

We evaluated the expression of PPARα and PPARα^SER12^ to determine if feno could stimulate their expression. As shown in the Figure 5, cells subjected to conditions of HR, HG, and HR/HG groups showed a decreased expression of PPARα (Figure 6A) and PPARα^SER12^ (Figure 6B), and the treatment with feno increased their expression, suggesting that feno is able to reach its target and activate it (Figure 6A,B). Because it was reported that the activation of PPARα generates the transcription of 14-3-3ɛ [13], we evaluated its expression and observed that the activation of PPARα promotes an increase in the expression of this protein (Figure 7A).

The apoptotic intrinsic pathway involves changes in the inner membrane of the mitochondria, opening the mitochondrial permeability transition pore and releasing apoptosis effectors such as cytochrome C which can bind to procaspase-9 to form the apoptosome. Therefore, we evaluated antiapoptotic proteins that include phosphor-Bad, 14-3-3-ε, and Bcl-2 (Figure 7). It is known that phosphor-Bad can be retained in the cytosol if it is bound to 14-3-3-ε, acting as antiapoptotic factors. We observed that the expressions of 14-3-3-ε (Figure 7A) and phosphor-Bad (Figure 7B) were decreased at HR, HG, and both HR/HG, but they were increased in the groups treated with feno. We also observed that unphosphorylated Bad (Figure 7B) did not change in any condition. Bcl-2, an antiapoptotic protein, exhibited a lower expression in HR, HG, and their combination. Meanwhile, feno treatment was able to raise Bcl-2 expression in HR, HG, and HR/HG groups (Figure 7C).

As expected, the pro-apoptotic mitochondrial protein Bax increased in the HR, HG, and HR/HG groups due to pathological conditions. Feno treatment brought down its expression in every condition (Figure 8A). There are initiator caspases, such as caspase-9, and executor caspases, such as caspase-3, that trigger the apoptosis process; upon activation, they promote the release of cytochrome C from the mitochondria to the cytosol. Accordingly, we observed that cells exposed to HR, HG, or HR/HG exhibited a raised expression of caspase-9 (Figure 8B) and cytoplasmic cytochrome C (Figure 8C). Cytoplasmic cytochrome C was not found in the mitochondria of cells treated with fenofibrate in HR, HG, or HR/HG. Treatment with the PPARα agonist increased the expression of cytochrome C in the mitochondria, suggesting a protector effect from apoptosis (Figure 8D).

### 2.7. Effect of Fenofibrate in Mitochondrial Potential Using Mitotraker (MTRK) Deep Red

Labeling of mitochondria in live and fixed cells with MitoTracker dyes was performed (Figure 9). This assay was used to mark mitochondria of live cells since its accumulation is dependent of transmembrane potential in mitochondria. Compared to control cells, the MTRK mark was lower in cells treated with HG and HR, and both conditions, the presence of feno in the pathological groups recovered membrane potential and therefore mitochondrial viability, observing an increase in the MTRK red color in cardiomyocytes.

### 2.8. Fenofibrate Treatment Attenuated the Damage Produced by HR, HG, and HR/HG to Ultrastructure in Mitochondria

To determine whether feno exerts any effect on myocardial histology, we evaluated the ultrastructure of mitochondria searching for apoptosis (Figure 10). We observed that mitochondria in the control group (Figure 10A) maintained their typical structure and an even distribution of mitochondrial crest. Cell cultures exposed to HG (Figure 10B), HR (Figure 10D), and HR/HG (Figure 10F) developed a less dense mitochondrial crest, the integrity of the crest was lost, and the mitochondria were elongated. When this situation exists, the outer and inner mitochondrial membranes lose the internal potential and release of apoptotic proteins from the mitochondria to the cytoplasm, then the mitochondria swells and cell death by apoptosis occurs. In contrast, treatment with feno attenuated the damage produced in the mitochondria. This was observed as reduced damage to the membranes, decreased density, and preservation of mitochondrial morphology, indicating that the mitochondria remain functional (Figure 10C, Figure 10E, and Figure 10G, respectively). Our data also show that feno attenuated the damage produced by HR, HG, and HR/HG in mitochondria.

## 3. Discussion

We evaluated the cellular protection generated by pharmacological preconditioning with fenofibrate in primary cultures of heart cells obtained from neonatal rats subjected to HR, HG, or HG/HR. In I/R injury, many pathophysiological processes occur, including accumulation of ions, damage to the mitochondrial membrane, formation of reactive oxygen species (ROS), alterations in nitric oxide (NO) metabolism, endothelial dysfunction, platelet aggregation, immune activation, apoptosis, and autophagy [14]. An essential characteristic of ischemia is that there is not enough oxygen to maintain oxidative phosphorylation by mitochondria and in the process of reperfusion structural, functional, and biochemical changes are caused in the myocardial tissue, which can result in cell death [15].

HIF-1α is a key regulator of HR-induced cardiomyocyte apoptosis and it increases apoptosis in primary cultures of heart cells obtained from neonatal rats. This activation could induce mechanisms that involve Bnip3 and caspase-3 [16]. In our research, we evaluated the expression of HIF-1α in primary cultures of heart cells subjected to HR and we observed an increase in the expression of this transcription factor, which is in accordance to what was reported (Figure 1), a decrease in cell viability (Figure 2), increased cytotoxicity (Figure 3A), and an increase in apoptosis and necrosis (Figure 4 and Figure 5).

Apoptosis is a type of cell death with characteristic alterations in cell morphology and cell fate which is different from death due to oncosis or necrosis. Apoptosis may be considered a mechanism that counterbalances the effect of cell proliferation by mitotic division in terms of tissue kinetics. In fact, deregulated apoptosis was implicated in the development a wide variety of human diseases. The interest currently aroused by the study of apoptosis in medicine is great since its dysregulation can contribute to the development of cardiovascular diseases [17]. Programmed cell death in the myocardium was linked to I/R injury as well as to excessive physical forces associated with increases in ventricular loading [18]. Cardiac cell death by HR due to apoptosis leads to loss of cardiac mass, decreased myocardial contractile capacity, and remodeling; apoptotic myocyte cell death precedes cell necrosis and is the major determinant of infarct size [17]. Apoptosis is a very important and definitive form of cellular death and has a close relation with cardiovascular diseases such as myocardial failure in contractile activity, arrhythmias, and cardiac remodeling.

Hyperglycemia is an important factor in cardiovascular damage [19], working through different mechanisms such as activation of protein kinase C, polyol hexosamine pathways, and advanced glycation end products. All of these pathways, in association to hyperglycemia-induced mitochondrial dysfunction and endoplasmic reticulum stress, promote ROS accumulation that, in turn, promotes cellular damage and contributes to the development and progression of diabetic complications. Hyperglycemia leads to an increase in oxidative stress and activates the calcium channels of cardiomyocytes that cause an acute rise in intracellular calcium concentration [20]. Intracellular calcium overload results in mitochondrial calcium accumulation which increases apoptosis and may explain the poor prognosis in diabetic patients after ischemia/reperfusion injury. Hyperglycemia is also an important risk factor for myocardial infarction. It makes cardiomyocytes prone to be damaged by hypoxia. Hyperglycemia and hypoxia are the main activators of apoptosis. According to Gan L et al. [21], hearts from diabetic subjects exhibit worse myocardial infarction/reperfusion-induced damage, increased infarct size, and increased apoptosis with no cardiac functional recovery.

Oxidative stress generated by excess ROS may play a role in the development and progression of cardiovascular disease. 8-hydroxy-2′-deoxyguanosine (8-OHdG) is a marker of oxidative DNA damage caused by ROS [22], and ROS generation induces malonate-mediated apoptosis through the proapoptotic Bcl-2 family protein Bax [23]. In our research, we observed that PPARα activation with feno reduces 8-OHdG and malonate concentrations in neonatal cardiomyocytes subjected to HR, HG, or HR/HG (Figure 3B,C). Bax expression is also significantly decreased in these cells (Figure 8A). PPARα activation results in binding to PPAR response elements (PPRE) in the 14-3-3ɛ promoter and upregulates 14-3-3ɛ transcription, increasing the expression of this protein and also increasing the interaction and sequestration of Bad. It therefore reduces Bad interference with the protective actions of Bcl-2 and Bcl-xl [13]. In addition, the PI3K/Akt signaling pathway promotes Bad phosphorylation, generating the interaction with 14-3-3ɛ but preventing Bad from inhibiting the antiapoptotic activity of Bcl-2 [24]. Some studies showed that PPARα activation promotes downstream PI3K/Akt activation [10]. We observed that pharmacological preconditioning with feno in primary cultures of cardiomyocytes obtained from neonatal rats subjected to HG, HR, or both conditions generated an increase in the expression and activity of PPARα (Figure 5A,B) and an increase in the expression of p-Bad (Figure 7B) and 14-3-3ɛ (Figure 7A), similar to what was reported.

Wang M et al., pointed out that PI3K/Akt signaling mediates cell survival and plays a vital role in the brain, intestine, and liver ischemia reperfusion injury, and is associated with apoptosis and the inflammatory response. PI3K, upon activation, transmits extracellular signals to Akt to promote cell survival by modulating the Bcl-2-associated death promoter (Bad). Phosphorylation of Bad by Akt suppresses caspase-3 activity and depolymerizes Bcl-2 to reduce cell apoptosis [25]. In previous studies conducted in our laboratory, we observed that PPARα activation with fenofibrate promotes an increase in the expression of the PI3K/Akt pathway in a model of metabolic syndrome and I/R [9]. We also observed that PPARα activation promotes an increase in the expression of p-Bad (Figure 7B) and Bcl-2 (Figure 7C), which would be associated with the decrease in the number of apoptotic cells (Figure 3 and Figure 4).

Pharmacological preconditioning with PPARα ligands, including fibrates, reduces myocardial I/R injury in non-diabetic and diabetic animals [11,26]. In our research, the activation of PPARα by feno in primary cultures of cardiomyocytes obtained from neonatal rats and subjected to HR, HG, or both conditions generated cellular protection observed as an increase in cell viability (Figure 2), a decrease in cytotoxicity (Figure 3A), and in the number of apoptotic cells (Figure 4 and Figure 5), similar to what was reported in the literature. In addition, feno increased mitochondrial viability (Figure 9) and improved the ultrastructure of these organelles (Figure 10).

During apoptosis, the integrity of the outer and inner mitochondrial membranes is altered, leading to the formation and opening of mitochondrial pores, the loss of inner potential, and the release of proapoptotic proteins into the cell cytoplasm. Upon apoptotic stimuli, the cardiomyocytes are activated and oligomerize at the mitochondrial outer membrane to mediate its permeabilization, which leads to the release of proapoptotic factors, such as cytochrome C (Cyt C) and subsequent initiation of the caspase cascade, which is considered a key step in apoptosis [27,28]. Our results show that Cyt C in cytoplasm increased in HR, HG, and both conditions, while the treatment with feno maintains Cyt C in the mitochondria, preventing the activation of apoptosis. This suggests cellular protection (Figure 8C,D). Taken together, our results suggest that the increased expression of Bcl-2 blocked the activation of Bax, inhibiting the release of Cyt C during HR, HG, and both conditions. This rescue activity is also correlated with inhibition of caspase-9 and -3 activation, thereby enhancing cell viability as reported by Reshi L et al. [29]. Similar results were observed by Kimura H et al., who treated murine renal proximal tubules with fenofibrate and observed a reduced release of Cyt C to cytoplasm compared to proximal tubular cells treated with cisplatin [30].

Mitochondria perform crucial roles in many biochemical processes; therefore, damaged mitochondria produce reactive ROS, inducing rapid depolarization of inner mitochondrial membrane potential (ΔΨm) with subsequent impairment of the respiratory chain to generate ATP, which is required for maintaining cardiomyocytes quality both in vitro and in vivo. Mitochondria perform crucial roles in many biochemical processes, and mitochondrial depolarization is an early sign of apoptosis. The decrease in ΔΨm simultaneously with the increase in MitoTracker expression indicates the occurrence of mitochondrial swelling, which reflects alterations in the internal mitochondrial membrane that gradually loses its crista ridges, causing volume expansion and loss of the capacity to generate energy [31,32]. This mitochondrial edema is required for the release of Cyt C during the intrinsic pathway of apoptosis [33]. MitoTracker is a commercially available fluorescent dye (Invitrogen/Molecular Probes) that labels mitochondria within live cells utilizing the mitochondrial membrane potential. Our results show that fenofibrate maintains membrane potential while HR, HG, and both pathologies reduce it (Figure 9). Our results are similar to those obtained by Kar D. et al. [34], where rat cardiomyocytes H9c2(2-1) and neonatal rat ventricular myocytes (NRVMs) were cultured and treated with α1-adrenergic agonist phenylephrine (PE, 100 µM, 24 h) in the presence or absence of 10 µM fenofibrate or bezafibrate. Mitochondrial transmembrane potential (Δψm) and motility were reduced by PE, which was significantly increased by fenofibrate.

As observed in Figure 10, the ultrastructure of mitochondria shows loss of cristae structure, expansion of the mitochondrial matrix, increased mitochondrial size, and hyperdensity of the mitochondrial matrix due to HR, HG, and HR/HG conditions. In general, proapoptotic proteins allow the opening of the permeability transition pore, a channel formed in the inner mitochondrial membrane letting out the Cyt C. This produces mitochondrial depolarization [35], while fenofibrate partly prevented the damage leading to mitochondrial conserved crests, which show almost normal size; there is a little fragmentation of the mitochondrial membrane and we observe that Cyt C is kept in the mitochondria since the expression of Cyt C in the mitochondria is increased in the treatment with Feno. Additionally, the presence of fenofibrate promotes a reduction in oxidative stress [8], inhibiting apoptosis and leading to shrinkage in mitochondria due to decreased ion exchange across the mitochondrial inner membrane, this ultimately protects both mitochondrial morphology and function.

## 4. Material and Methods

### 4.1. Animals

Newborn male and female Wistar rats (1–3 days), provided by the animal facilities of the Center for Research and Advanced Studies of the National Polytechnic Institute (CINVESTAV, IPN) were used to obtain primary neonatal rat heart cells. The protocol was carried out following the guidelines of the Institutional Ethics Committee (Protocol Number 0270/18). All the experiments were conducted in accordance with Institutional Ethical Guidelines (Ministry of Agriculture, SAGARPA, NOM 062-ZOO-1999, Mexico).

### 4.2. Neonatal Rat Cardiomyocytes (NRCMs) Isolation and Culture

Neonatal rat cardiomyocytes were isolated and cultured following a previously described methodology (8). The experiments were carried out in a monolayer of cardiomyocytes obtained on the 3rd or 5th day of culture. Cells were subdivided into the following experimental groups: Group 1—control (Ctrl), Group 2—subjected to hypoxia/reoxygenation (HR), Group 3—HR + 10 μM fenofibrate (feno) (Sigma-Aldrich, St. Louis, MO, USA), Group 4—High glucose (HG, 25 mM glucose; Gibco, Waltham, MA, USA), Group 5—HG + feno, Group 6—HR/HG, and Group 7—HR/HG + feno. To evaluate viability and hyperosmolarity, the following groups were also studied: Ctrl + 0.1% dimethyl sulfoxide (DMSO) (vehicle for fenofibrate) and Ctrl + 19.5 mM D-mannitol (Sigma-Aldrich, St. Louis, MO, USA). Cardiomyocytes were exposed to HG for 48 h in F-10 medium (Sigma-Aldrich, St. Louis, MO, USA) containing 25 mM glucose. To produce hypoxia in cultured cardiomyiocytes, we used anaerobic bags (GasPack TM EZ system, BD Biosciences, USA) at 37 °C in standard incubator [36,37]. Briefly, six well plates containing cultured cardiomyocytes were exposed, for 2 h, to an atmosphere composed by 95% N2/5% CO_2_ into a sealed bag containing an oxygen-consuming palladium catalyst, resulting in a hypoxic environment (25–35 mmHg PO_2_). Immediately after the hypoxia period, cell cultured plates were placed in a standard incubator for reoxygenation for 1 h before further assays [38,39]. The success to produce hypoxia was evaluated through the increased expression of HIF-1α by Western blot [36]. The experimental groups are represented in the Figure 11. Mannitol (19.5 mM) and high glucose (25 mM) were administered 48 h before fenofibrate (10 μM) or DMSO (0.1%). The treatment with fenofibrate or DMSO lasted for 4 h.

### 4.3. Cell Viability

Cell viability was evaluated according to previously reported method by Cortes-López et al. [8]. After HR, cells were stained with 100 μL trypan blue dye (0.4%) and an aliquot of 50 µL was placed in a Neubauer chamber (Neubauer, Marienfeld, 0.0025 mm^2^; Wollerspfad Lauda Konighofen, Germany). Blue-stained cells (death) and non-dyed cells (alive) were counted under a microscope at 10× magnification. The cell viability must be at least 95% to consider a viable culture. The study was repeated 4 times and data were combined and analyzed for statistical significance.

### 4.4. Quantification of Cytotoxicity

To determine cytotoxicity under our experimental conditions, the MITO-ID membrane potential (MP) cytotoxicity kit was used (Enzo Life Sciences, NY, USA). Briefly, a culture plate containing approximately 2 × 10^4^ to 3 × 10^4^ cells per well were carefully removed from the culture medium, placing the respective cell button in each well of a 96-well fluorescence plate. Fifty μL of isotonic saline solution at 37 °C were added and gently homogenized for 1 min. Freshly prepared Mito-ID MP reagent (100 μL) was added and gently homogenized for 1 min and incubated at room temperature for 30 min protected from light. Then, 100 μL of buffer 1 were added and gently homogenized for 1 min. Additionally, 20 μL of the carbonyl cyanide 3-chlorofenylhydrazone (CCCP) reagent were prepared in buffer 2 at a concentration of 2 to 4 μM and were added and gently homogenized. Fluorescence was monitored (disappearance of the orange color) at an excitation length of 490 nm and an emission length of 590 nm for 15 min. The control sample was evaluated without CCCP reagent. A greater potential for cytotoxicity of the mitochondrial membrane generated greater consumption of Mito-ID membrane potential and therefore less intensity in the orange color of the reaction. It is expressed as relative fluorescence units (RFU) of Mito-ID MP consumed during 15 min [40].

### 4.5. Quantification of 8-hydroxy-2-deoxyguanosine (8-OH-2dG)

8-HO-2dG was determined in 1 × 10^6^ cultured cells of each experimental group by capillary zone electrophoresis, using UV detection and diode array, at 20 kV, for 8 min, at 200 nm according to the methodologies of Kvasnicová and Tuma [41,42]. Results are expressed in pmoles/mL.

### 4.6. Quantification of Malonate (MTO)

Malonate (MTO), a metabolite considered a damage marker in an apoptotic process, was determined in the cell culture (1 × 10^8^ cells) of each experimental group by capillary zone electrophoresis. The sample was deproteinized with cold methanol (1:1 ratio), centrifuged at 16,000× *g* for 15 min (Spectrafuge 24D, Labnet, Co., Urbana, IL, USA), and filtered with 0.22 μm nitrocellulose membrane filters (Millipore, Billerica, MA, USA), diluted 1:3 with cold sodium hydroxide 0.1 M, and analyzed with a Beckman Coulter P/ACE TM MDQ system. The capillary was preconditioned with sodium hydroxide solution (0.1 M) at 20 psi for 10 min, then distilled in water for 10 min and finally the running buffer (10 mM borates + 0.5 mM sodium citrate), pH 9.0, for 10 min. Samples were injected under hydrodynamic pressure at 0.5 psi/10 s. The separation was carried out at −25 KV for 4 min at 267 nm. The capillary was washed between runs at 20 psi with NaOH (0.1 M) for 2 min, deionized water for 2 min, and running buffer for 4 min. The concentration of MTO is expressed in pmoles/mL and was determined by means of a standard curve.

### 4.7. Subcellular Fractioning

To determine the damage to organelles in the cell affected by apoptosis, we evaluated by Western blot a subcellular fractioning (nucleus, mitochondria, and cytosolic fraction). Scraped cells were washed with cold PBS and pelleted by centrifugation at 200× *g* for 10 min. Every sample was resuspended in 500 μL fractionation buffer (20 mM Hepes pH 7.4, 10 mM KCl, 2 mM MgCl_2_, 1 mM EDTA, 1 mM EGTA, and protease- and phosphatase- inhibitor cocktails), and homogenized for 1 min on ice using a tight-fitting Teflon pestle and a Potter Elvehjem homogenizer. The homogenate was kept on ice for 20 min and then centrifuged at 720× *g* for 5 min. The pellet (P1) was labeled as nuclear fraction and kept on ice or frozen. The supernatant (S1) was decanted into a centrifuge tube and centrifuged at 10,000× *g* for 5 min. The pellet (P2) containing mitochondrial fraction was maintained on ice or frozen, while the supernatant (S2) containing cytosolic fraction was kept on ice. Cytosolic proteins were precipitated from the supernatant (S2) in cold 12% trichloroacetic acid, incubated overnight at 4 °C, and centrifuged at 720× *g* for 10 min. Once centrifuged, the supernatant (S3) was discarded and the pellet (P3) containing the cytosolic fraction was frozen at −70 °C until use. All buffers and centrifugation steps were modified from Dimauro et al. [43].

### 4.8. Protein Expression by Western Blot

Total protein was quantified by BCA Protein Assay Kit (Pierce, Waltham, MA, USA) as previously reported by Cortés-López et al. [8]. Eighty μg of protein were electrophoretically separated on a 12% SDS-PAGE gel to 100 V for 2 h, followed by electrotransference to a polyvinylidene fluoride (PVDF) membrane (0.45 μM Millipore, Billerica, MA, USA) at 10 V for 1 h. Membranes were incubated with 5% nonfat milk (Bio-Rad, Hercules, CA, USA). Blots were probed with specific antibodies against HIF-1α (1:250), 14-3-3ε (1:250), total Bad (1:100), p-Bad (1:100) (Santa Cruz Biotechnology Inc., Santa Cruz, CA, USA), Bcl-2 (1:1000) (InVitrogen, Camarillo, CA, USA), caspase-9 (1:300), Bax (1:50) (Bio- Vision, Milpitas, CA, USA), cytochrome C (1:50) (Sigma-Aldrich, Saint Louis, MO, USA), PPARα (1:100), and p-PPARα (1:100) (Abcam, Cambridge, UK). Blots were stripped and reincubated with β-actin (1:5000) or VDAC (1:500) (mitochondria marker) (Santa Cruz Biotechnology Inc., Santa Cruz, CA, USA), and these antibodies were used as protein load control. Signals were detected by Immobilon Western Chemiluminescent HPR substrate (Millipore, Billerica, MA, USA). Images from each film were acquired by a GS-800 densitometer (including 29.0 version Quantity One software from Bio-Rad Laboratories, Inc., Hercules, CA, USA). Values of each band density are expressed as arbitrary units.

### 4.9. TUNEL for Apoptosis Detection

Cell death by apoptosis was detected in neonatal cardiomyiocytes cells from the primary culture of all experimental groups using the TUNNEL assay. Once cultured cells reached confluence, the culture dishes were placed out of the incubators until they reached room temperature, and then the culture medium was removed. The slides with the adhered cells were placed in PBS with 10 mM CaCl_2_ and 5 mM MgCl_2_ and washed 3 times (3 × 5 min) with the same solution. They were fixed with 4% PFA (4–8 °C) for 10 min, followed by 3 washes with PBS (3 × 5 min). Cells were permeabilized with 0.1% Triton X-100/0.1% sodium citrate for 2 min at 4 °C. The area for the reaction was delimited within the slide with a hydrophobic pencil, and they were incubated with the TUNEL reaction mix [(TdT-mediated dUTP nick end labelling) (ROCHE-11 684 795 910, In Situ Cell Death Detection Kit, Fluorescein)] for 60 min at 37 °C in a humid chamber and in the dark. An equivalent area was included as a positive control (treated with DNAse) and another as a negative control (with the labeling buffer, without the enzyme). At the end of the incubation, they were brought to room temperature, washed 3 times (3 × 5 min) with PBS, and the nuclei were labeled with 4′,6-diamidino-2-phenylindole (DAPI) for subsequent microscopic observation. In total, 4 to 6 photographs were acquired at 20× on a fluorescence microscope (Floid Cell Imaging Station, Life Technologies). The total number of nuclei (blue + green) was quantified and the % of cells in apoptosis (green) was calculated, with a total of 2 to 4 thousand cells per condition [44].

### 4.10. Determination of Apoptosis, Necrosis, and Dead Cells by Annexin V/Propidium Iodide (PI) Staining

This assay was used to measure the number of cells that underwent apoptosis, necrosis, and dead cells. In short, 1 × 10^6^ cells were seeded in 6-well plates and treated as follows: control (Ctrl), cells with 25 mM glucose (HG), cells with 10 μM fenofibrate/25 mM glucose (HG-Feno), cells in hypoxia/reoxygenation (HR), cells in hypoxia/reoxygenation with 10 μM fenofibrate (HR-Feno), cells in hypoxia/reoxygenation with 25 mM glucose (HR/HG), and cells in hypoxia/reoxygenation with 25 mM glucose + 10 μM fenofibrate (HR/HG-Feno). Cells were detached by trypsin and put in a 1.5 mL Eppendorf tube, washed with 1 mL ice-cold phosphate buffer, and centrifuged at 1500 rpm for 4 min, supernatant was discarded and cells were resuspended in 100 μL Annexin binding buffer (10 mM HEPES, 140 mM NaCl, 2.5 mM CaCl_2_ in dH_2_O, pH 7.4), and 5 μL of FITC-Annexin V and 1 μL propidium iodide (100 mg/mL) were added per tube and incubated for 15 min at room temperatura; 400 μL of Annexin binding buffer were further added into the tube after incubation. The stained cells were analyzed by flow cytometry (Cytometer FACS Aria Fusion) including 5000 events per experiment. This assay was repeated three times [44].

### 4.11. Cell Loading with MitoTracker Deep Red

MitoTracker is a commercially available fluorescent dye that labels mitochondria within live cells, utilizing the mitochondrial membrane potential. As MitoTracker is chemically reactive when linking to thiol groups in the mitochondria, the dye becomes permanently bound to the mitochondria, and thus remains after the cell dies or is fixed. In addition, it can be used in experiments in which multiple labeling diminishes mitochondrial function. In short, 1 × 10^5^ cells of each experimental group were seeded in 6-well plates including circular coverslips. Media was removed from each well with grow cells on coverslips and washed two times with PBS buffer, then they were incubated for 45 min with 250 μM of Mitotracker Deep Red solution in dark; after staining, the solution was replace with fresh buffer, and the buffer was replaced with 4% paraformaldehyde and incubated 15 min to fix. After incubation, the paraformaldehyde was removed and samples were washed twice with PBS and incubated for 1 h with DAPI solution. Finally, coverslips were mounted in slides using mounting solution (Mowiol) and observed using a fluorescence microscope Floid Cell Imaging Station (Life Technologies) [45].

The images were acquired with a Floid Cell Imaging Station (Life Technologies) and the quantification of the integrated optical density [IOD (lum/pix^2 × 10^6^)] was carried out with the Media Cybernetics software, Images-Pro Premier 9. Eight 20× images (4/well) of cells were seeded in coverslips and treated as mentioned above [46].

### 4.12. Electron Microscopy

Samples of all experimental groups were processed as described by Gónzalez-Morán et al. We used 1 × 10^6^ cells. Ultrathin sections (approximately 60 nm thick) were obtained using a Leica Ultracut microtome and subsequently mounted on copper grids and counterstained with uranyl acetate. The evaluation was carried out with a JEM-1011 (JEOL Ltd., Tokyo, Japan) at 60 kV and 60,000× [47,48].

### 4.13. Data Analysis

Results are expressed as the mean of 6 different experiments ± standard error of the mean. For multiple comparisons, we applied one-way or two-way analysis of variance (ANOVA) followed by Tukey’s post hoc test. For comparisons between two groups, an unpaired Student’s *t*-test was used. A *p* < 0.05 was considered a statistical difference.

## 5. Conclusions

Pharmacological preconditioning with fenofibrate improved culture of neonatal cardiomyocytes viability, decreased apoptotic cells through increasing anti-apoptotic proteins expression, and protected the mitochondria from damage induced by the pathological conditions installed by HR, HG, and HR/HG.

## Figures and Tables

**Figure 1 ijms-25-11391-f001:**
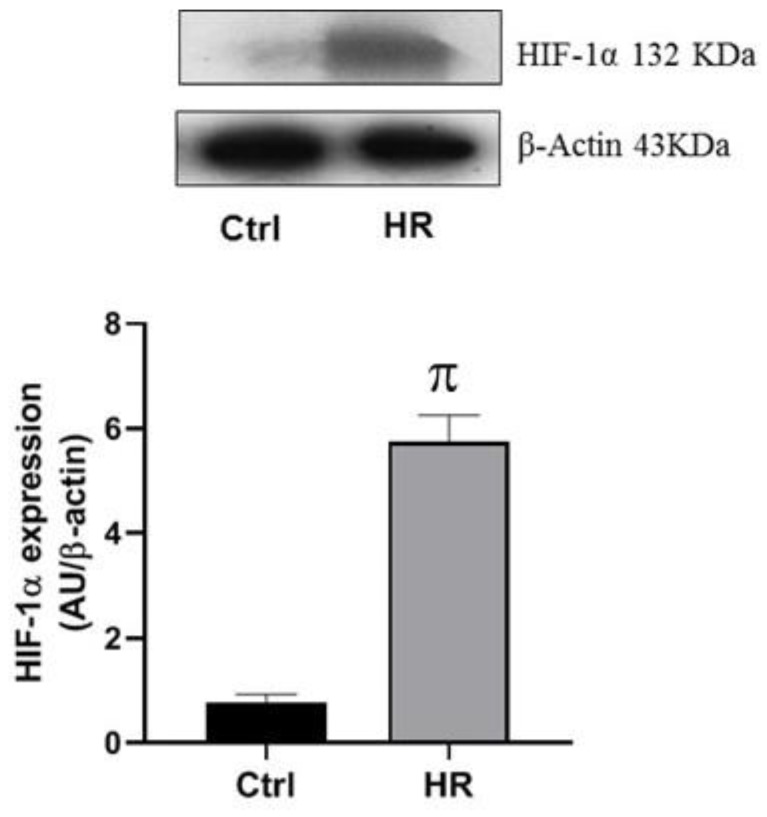
Representative Western blot and densitometric analysis of Western blot for the expression of HIF-1α. The values represent the mean ± standard error of the mean (SEM) of five different experiments ^π^ *p* < 0.05 vs. Ctrl, t-Student. HIF-1α, hypoxia induced factor 1 alpha; Ctrl, control group; and HR, hypoxia/reoxygenation group.

**Figure 2 ijms-25-11391-f002:**
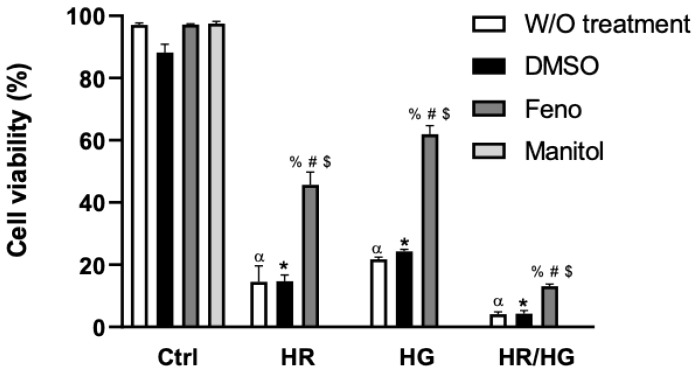
Effect of fenofibrate (10 μM) on cell viability. Cell viability was measured by trypan blue dye caption in cells from every experimental group: control (Ctrl), hypoxia/reoxygenation (HR), high glucose (HG), and a combination of both conditions (HR/HG). For the control group, an osmolarity control group was included with mannitol. Two-way ANOVA followed by Tukey’s post hoc test. n = 6 ^α^
*p* < 0.05 vs. Ctrl-W/O, * *p* < 0.05 vs. Ctrl-DMSO group, ^%^ *p* < 0.05 vs. Ctrl-Feno group, ^#^ *p* < 0.05 vs. W/O group, and ^$^ *p* < 0.05 vs. DMSO group.

**Figure 3 ijms-25-11391-f003:**
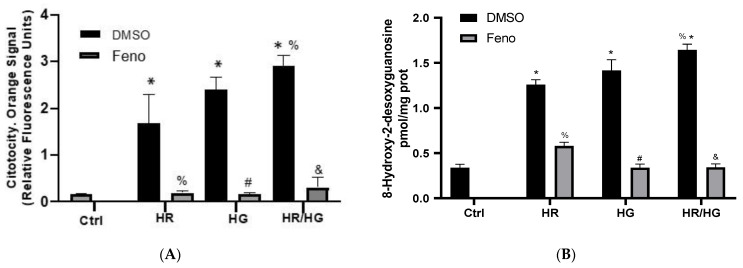
Fenofibrate prevents nuclear and mitochondria damage. (**A**) Cellular damage secondary to HR, HG, and HR/HG is prevented by feno (10 μM). (**B**) Nuclear damage induced by HR, HG, and HR/HG is prevented by feno. (**C**) Mitochondria respiration damage induced by HR, HG, and HR/HG is prevented by feno. One-way ANOVA followed by Tukey’s post hoc test. n = 6 * *p* < 0.05 vs. DMSO Ctrl, ^%^ *p* < 0.05 vs. DMSO HR group, ^#^ *p* < 0.05 vs. DMSO HG group, and ^&^ *p* < 0.05 vs. DMSO HR/HG group. Feno: fenofibrate treatment, Ctrl: control group, HR: hypoxia/reoxygenation group, HG: high glucose group, and HR/HG: hypoxia/reoxygenation and high glucose group.

**Figure 4 ijms-25-11391-f004:**
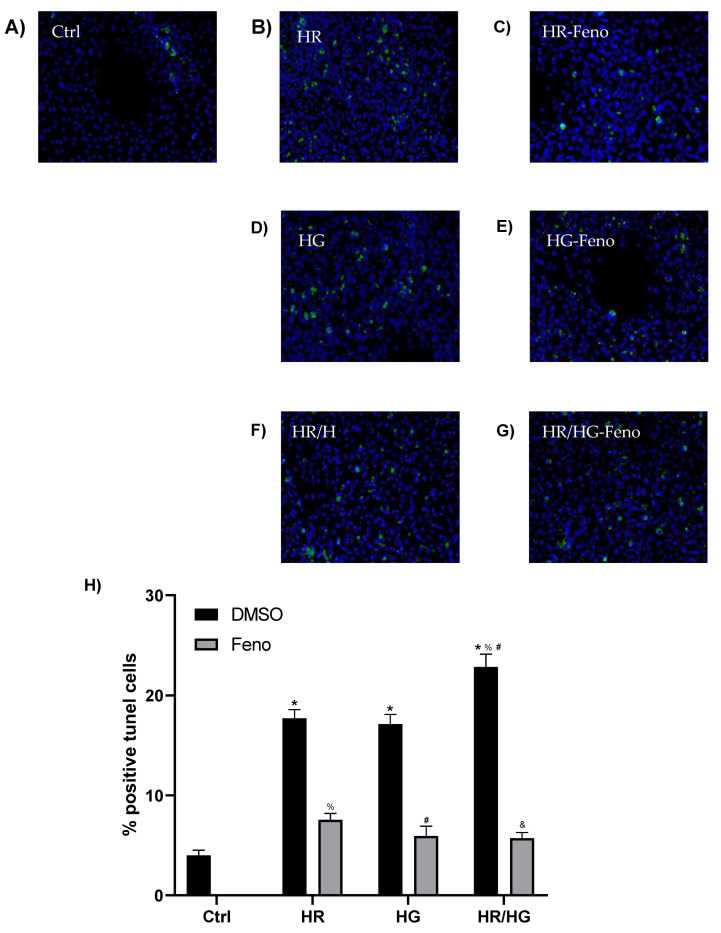
TUNEL test representative images of (**A**) Ctrl, (**B**) HR, (**C**) HR-F, (**D**) HG, (**E**) HG-F, (**F**) HR/HG, and (**G**) HR/HG-F. The green mark corresponding to cell nuclei in apoptosis and the blue mark for the other nuclei are observed. (**H**) Quantification of tunel images. One-way ANOVA followed by Tukey’s post hoc test, n = 6. * *p* < 0.05 vs. Ctrl-DMSO, ^%^ *p* < 0.05 vs. HR-DMSO group, ^#^ *p* < 0.05 vs. HG-DMSO group, and ^&^ *p* < 0.05 vs. HR/HG-DMSO group. Feno: fenofibrate treatment, Ctrl: control group, HR: hypoxia/reoxygenation group, HG: high glucose group, and HR/HG: hypoxia/reoxygenation and high glucose group.

**Figure 5 ijms-25-11391-f005:**
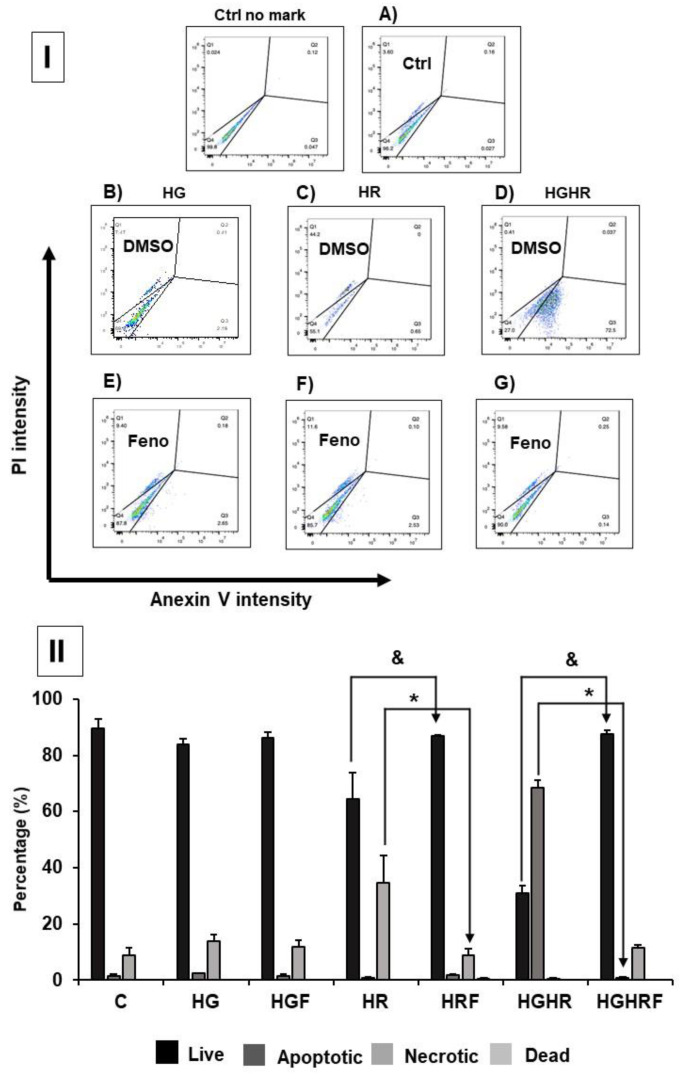
Annexin V staining of neonatal cardiomyocytes under HR, HG, and both HR/HG conditions with DMSO or fenofibrate treatment (**A**–**G**). (**I**) Images of PI and Annexin V at different conditions; Feno: fenofibrate treatment, Ctrl: control group, HR: hypoxia/reoxygenation group, HG: high glucose group, and HR/HG: hypoxia/reoxygenation and high glucose group. (**II**) Graph of percentage of intensity of fluorescence in each condition, showing different states of cells; Live, Apoptotic, Necrotic, and Dead. C: control, HG: high glucose, HGF: high glucose plus fenofibrate, HR: hypoxia/reoxygenation, HRF: hypoxia/reoxygenation plus fenofibrate, HGHR: hypoxia/reoxygenation and high glucose, and HGHRF: hypoxia/reoxygenation and high glucose plus fenofibrate. T-Student, ^&^ and * *p* < 0.05.

**Figure 6 ijms-25-11391-f006:**
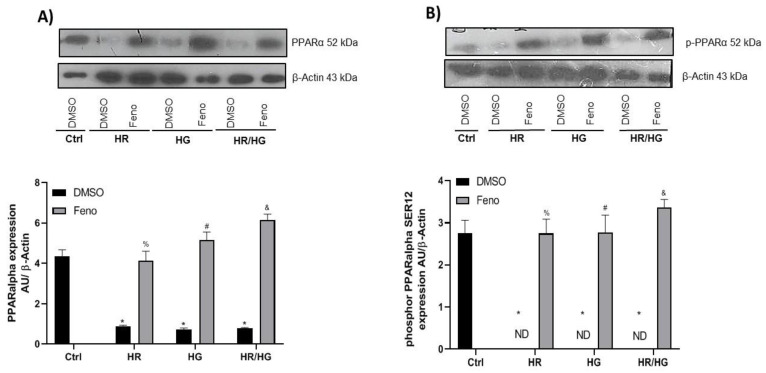
Expression and densitometric analysis of (**A**) PPARα and (**B**) PPARα^SER12^ proteins. One-way ANOVA followed by Tukey’s post hoc test, n = 6. * *p* < 0.05 vs. DMSO Ctrl, ^%^ *p* < 0.05 vs. DMSO HR group, ^#^ *p* < 0.05 vs. DMSO HG group, and ^&^ *p* < 0.05 vs. DMSO HR/HG group. Feno: fenofibrate treatment, Ctrl: control group, HR: hypoxia/reoxygenation group, HG: high glucose group, HR/HG: hypoxia/reoxygenation and high glucose group, and ND: not detected.

**Figure 7 ijms-25-11391-f007:**
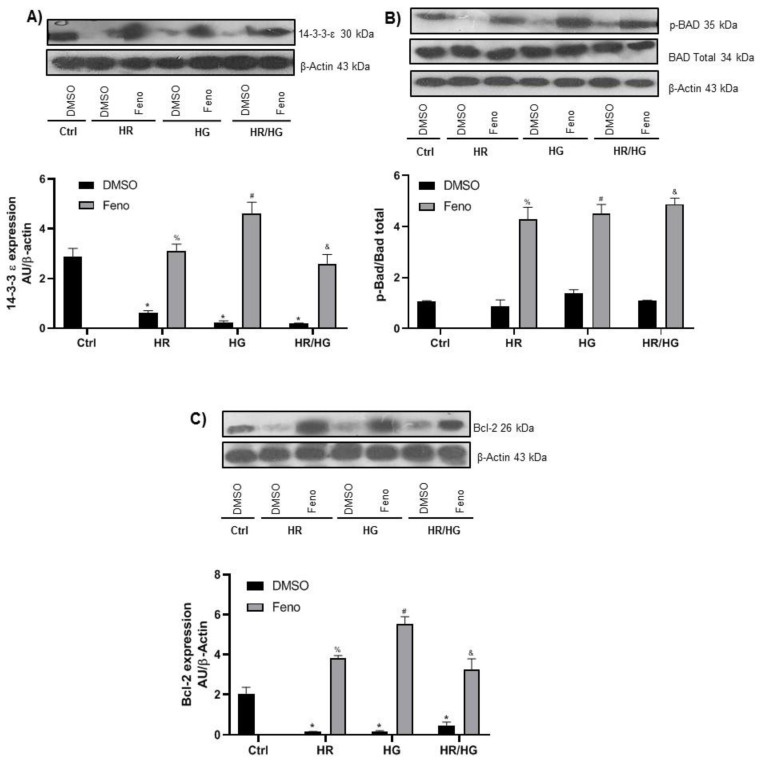
Expression and densitometric analysis of (**A**) 14-3-3-ε, (**B**) p-Bad, and (**C**) Bcl-2 proteins. One-way ANOVA followed by Tukey’s post hoc test, n = 6. * *p* < 0.05 vs. Ctrl-DMSO, ^%^ *p* < 0.05 vs. HR-DMSO group, ^#^ *p* < 0.05 vs. HG-DMSO group, and ^&^ *p* < 0.05 vs. HR/HG-DMSO group. Feno: fenofibrate treatment, Ctrl: control group, HR: hypoxia/reoxygenation group, HG: high glucose group, and HR/HG: hypoxia/reoxygenation and high glucose group.

**Figure 8 ijms-25-11391-f008:**
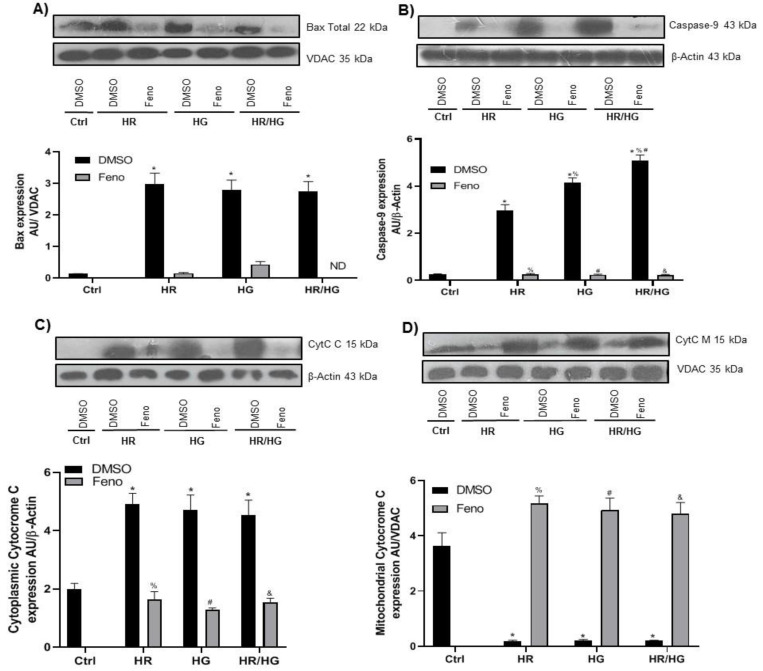
Protein expression and densitometric analysis of (**A**) Bax, (**B**) caspase-9, (**C**) cytoplasmic cytochrome c, and (**D**) mitochondrial cytochrome c proteins. One-way ANOVA followed by Tukey’s post hoc test, n = 6. * *p* < 0.05 vs. Ctrl-DMSO, ^%^ *p* < 0.05 vs. DMSO HR group, ^#^ *p* < 0.05 vs. DMSO HG group, and ^&^ *p* < 0.05 vs. DMSO HR/HG group. Feno: fenofibrate treatment, Ctrl: control group, HR: hypoxia/reoxygenation group, HG: high glucose group, and HR/HG: hypoxia/reoxygenation and high glucose group, and ND: not detected.

**Figure 9 ijms-25-11391-f009:**
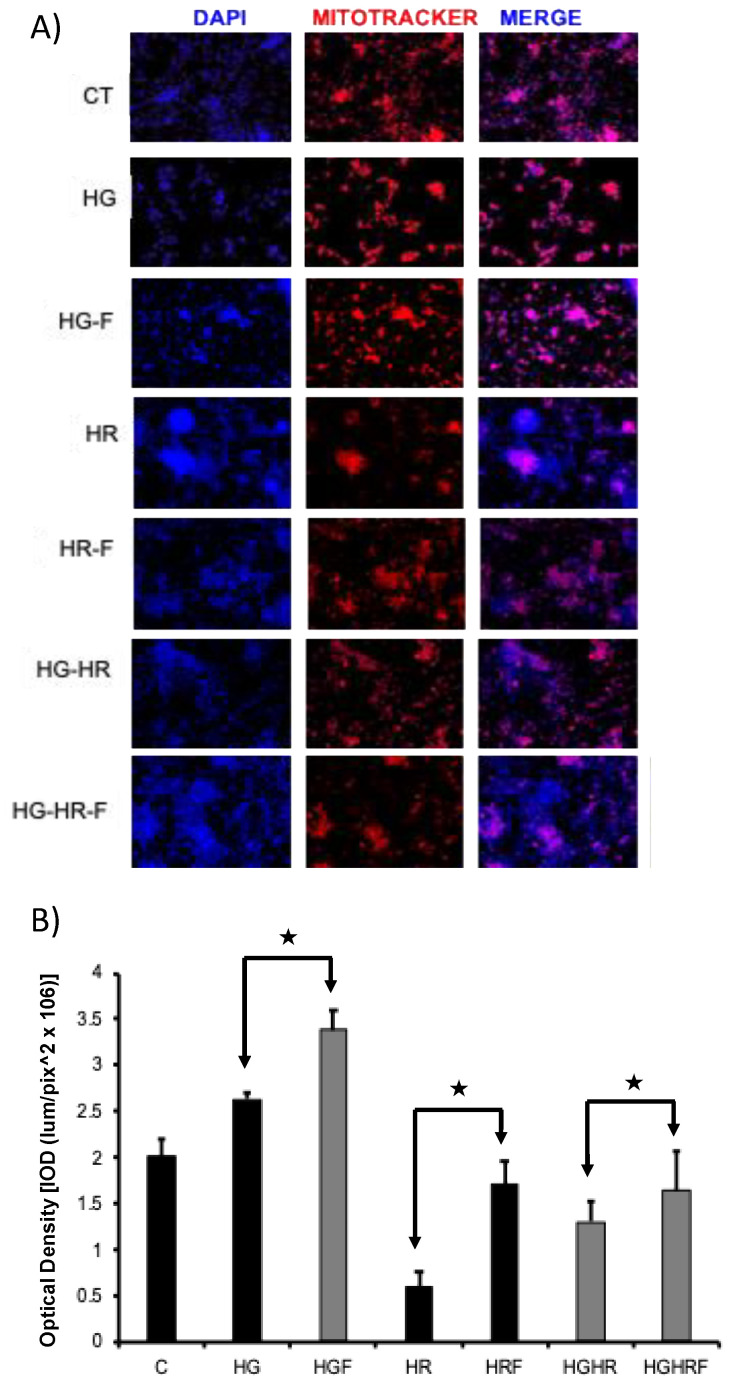
MitoTracker Deep Red staining in primary culture of neonatal cardiomyocyte cells. (**A**) Fluorescence microscopy images show the distribution of the Deep Red MitoTracker mitochondria-specific marker. (**B**) Eight 20× images (4/well) of each condition were quantified; IOD = (lum/pix^2 × 10^6^)]. The nuclei were labeled in blue, with 4′,6-diamidino-2-phenylindole (DAPI). Merge was carried out with a gray background to highlight the red MitoTraker mark. Feno: fenofibrate treatment, Ctrl: control group, HG: high glucose group, HR: hypoxia/reoxygenation group, and HR/HG: hypoxia/reoxygenation and high glucose group. ★ *p* < 0.05.

**Figure 10 ijms-25-11391-f010:**
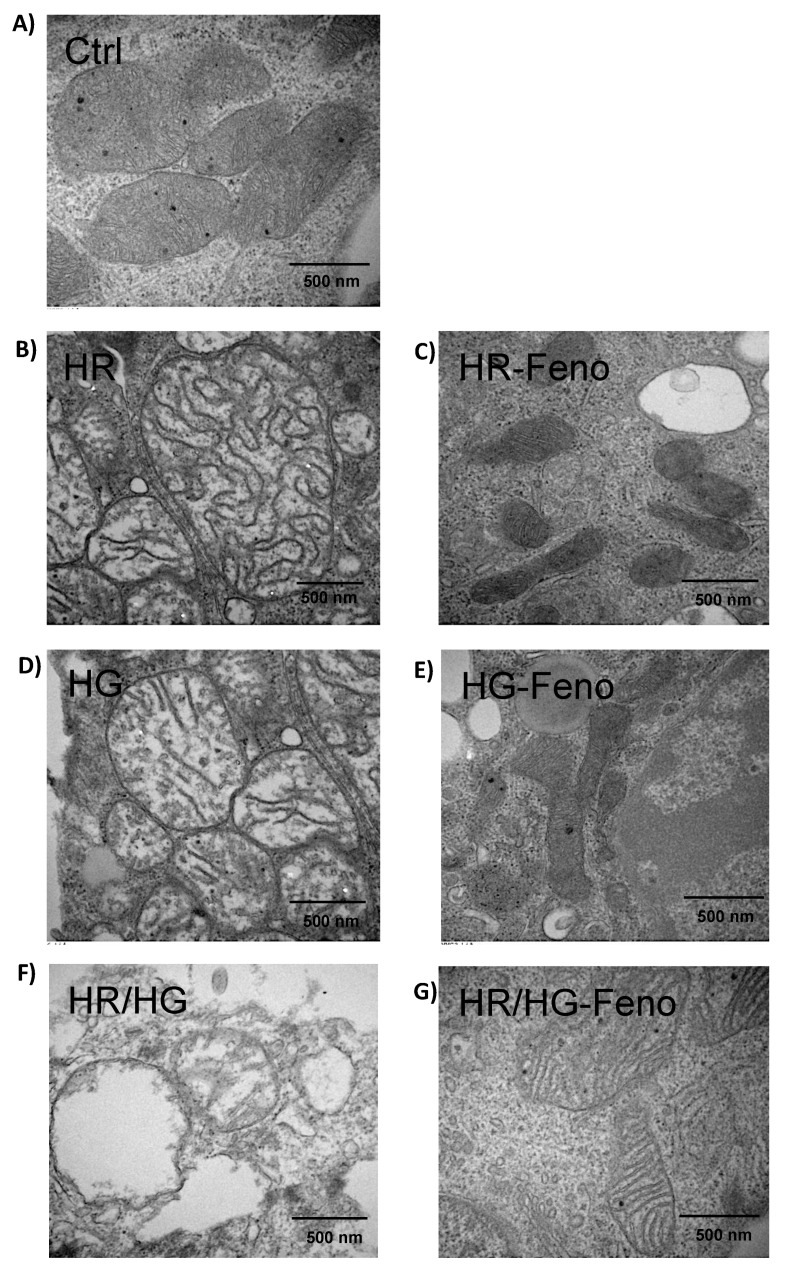
Ultrastructure of cardiomyocytes in primary cultures to observe mitochondria. (**A**) control, (**B**) HR, (**C**) HR-Feno, (**D**) HG, (**E**) HG-Feno, (**F**) HR/HG, and (**G**) HR/HG-Feno. Details of the ultrastructure by electron microscopy: 500 nm, 50,000×. Feno: fenofibrate treatment, Ctrl: control group, HR: reoxygenation group, HG: high glucose group, and HR/HG: reoxygenation and high glucose group. The images are representative of six experiments per group.

**Figure 11 ijms-25-11391-f011:**
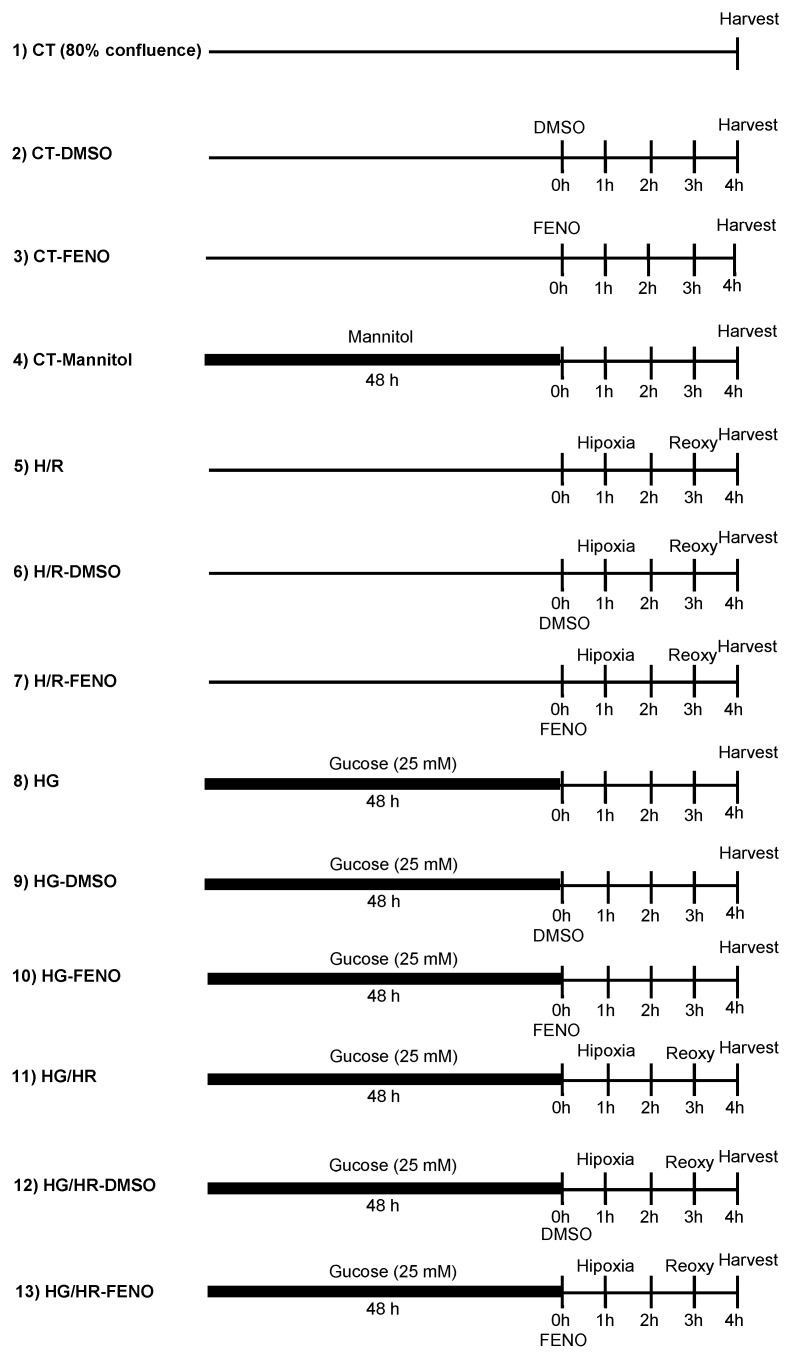
The experimental groups are represented in this figure. Mannitol (19.5 mM) and high glucose (25 mM) were administered 48 h before fenofibrate (10 μM) or DMSO (0.1%). The treatment with fenofibrate or DMSO lasted for 4 h; CT = control, HR = hypoxia/reoxygenation, DMSO = dimethylsulfoxide (0.1%), HG = high glucose, and FENO = fenofibrate (10 μM).

## Data Availability

The authors confirm that the data supporting the findings of this study are available within the article. The datasets of this study are available from corresponding author upon reasonable request.

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
