# Peer review of "Pharmacological Preconditioning with Fenofibrate in Cardiomyocyte Cultures of Neonatal Rats Subjected to Hypoxia/Reoxygenation, High Glucose, and Their Combination"

_ijms, 2024, doi:10.3390/ijms252111391_

Round 1

Reviewer 1 Report

Comments and Suggestions for Authors

In this manuscript the authors aim to show that pharmacological preconditioning with the fenofibrate (Feno)  WY-14643 is protective against ischaemia/reperfusion injury. This work follows on from a previous paper (ref 8) where they show that fenofibrate protects cardiomyocytes against hypoxia/reperfusion (HR) and/or high glucose (HG). In both papers neonatal rat cardiomyocytes were treated with normal media or mannitol as controls or with 25 mM glucose for 48 hours. In the previous paper cells were then treated with Feno for four hours. After the first hour cell cultures were covered with a coverslip for two hours and then the coverslip was removed and the cells were reoxygenated for 1 hour. In the current manuscript hypoxia was created using anaerobic bags for 2 hours and then cells were reoxygenated in the incubator. In the methods it states that cells were treated with Feno or DMSO for 4 hours but it is not clear from the methods or the figure of experimental groups that the Feno is applied for an hour before hypoxia. I assume it was or there wouldn’t be ‘preconditioning’ but the authors need to make this clear.

In both papers cells were treated with 25 mM glucose with is well above physiological levels and so it is hard to determine whether detrimental effects result from physiological effects of high glucose or from glucotoxicity.

In the manuscript the authors show that HG causes some necrosis but this is not affected by Feno, HR causes necrosis which is reduced by Feno and HR/GG causes very high levels of apoptosis which is eliminated by Feno. They show that PPARa is activated by Feno(as shown in ref 8), as were anti-apoptotic factors and that pro-apoptotic Bax was increased in HR, HG and HR/HG, which is interesting since Fig 6 showed that apoptosis was only detected in HG/HR. The authors then aim to show in Figures 10 and 11 that HG, HR and HR/HG caused mitochondrial damage but this is not particularly clear.

In summary, this paper shows that fenofibrate activates PPARa can be protective against extreme ischaemia and high glucose in isolated cardiomyocytes. They claim this is due to anti-apoptotic factors but they only show that apoptosis is elevated in HR/HG. Furthermore, this has been shown in various animal models and so is not particularly novel.

Points to clarify

The authors show that HIF is present at the end of this protocol which is interesting since HIF protein is expected to be downregulated rapidly on re-oxygenation and suggests that oxygen is not delivered back to the cells through the cell culture media very efficiently

Figure 3 shows that HR, HG and HR/HG reduce cell viability (as shown previously in ref 8) but there are no indicators of significant differences from control cells in the cells without treatment

Figure 4 shows the cells suffer nuclear and mitochondrial damage in HR/HG and HR/HG and this is rescued by Feno. At the start of paragraph 3.3 it says that the hallmark process of cytotoxicity is the secretion of granules but the methods say that cytotoxicity is measured via mitochondrial membrane potential with MITO-ID. The data in Fig 4A is presented as pmol/mg prot so it is not clear exactly what is shown.

Figure 10 shows labelling of mitochondria in ‘live and fixed cells’ – not sure what that means – and is suggested to show that this live mitochondria were reduced by HR, HG and HR/HG and increased using Feno but this is not clear from the images provided and should be quantified. Figure 11 (labelled as Figure 10) shows that treatment with HR, HG and HR/HG damages mitochondria and that Feno is protective but the images of HR-Feno and HG-Feno-treated mitochondria seem a very different size from controls or treated mitochondria.

Some aspects of the manuscript are over-detailed, I suspect they have been taken from a PhD thesis and could benefit from further editing. For example, in some methods (eg 2.10 and 2.11) unnecessary detail is provided

Comments on the Quality of English Language

The English is OK in most places but occasionally needs minor correction to clarify what ahs been done or seen.

Author Response

Thank you very much for your comments and suggestions.

Comments and Suggestions for Authors

  1. In this manuscript the authors aim to show that pharmacological preconditioning with the fenofibrate (Feno) WY-14643 is protective against ischemia/reperfusion injury. This work follows on from a previous paper (ref 8) where they show that fenofibrate protects cardiomyocytes against hypoxia/reperfusion (HR) and/or high glucose (HG). In both papers neonatal rat cardiomyocytes were treated with normal media or mannitol as controls or with 25 mM glucose for 48 hours. In the previous paper cells were then treated with Feno for four hours. After the first hour cell cultures were covered with a coverslip for two hours and then the coverslip was removed and the cells were reoxygenated for 1 hour. In the current manuscript hypoxia was created using anaerobic bags for 2 hours and then cells were reoxygenated in the incubator. In the methods it states that cells were treated with Feno or DMSO for 4 hours but it is not clear from the methods or the figure of experimental groups that the Feno is applied for an hour before hypoxia. I assume it was or there wouldn’t be ‘preconditioning’ but the authors need to make this clear.

ANSWER

In this study we are NOT using the drug WY-14643, nor are we studying its effect. We are studying the effect of FENOFIBRATE. It is well known that both are specific activators of PPARa but they correspond to very different molecules and their structure varies considerably:

WY-14643                                             FENOFIBRATE

As you comment, the methods indicated that the cells were treated with FENO or DMSO for 4 hours, but in the Figure of the Experimental Groups this is not observed. The reality is that FENO was added one hour before the HYPOXIA as you comment. We apologize because we made a mistake at the moment of including the insert in the Figure. We also want to mention that there effectively exists a pharmacological preconditioning with FENO, because it was incubated one hour before subjecting the cultures to hypoxia. We indicate with yellow in the Figure description the change made in the Image.

  1. In both papers cells were treated with 25 mM glucose with is well above physiological levels and so it is hard to determine whether detrimental effects result from physiological effects of high glucose or from glucotoxicity.

Answer

We agree with your comment, the isolated cells were exposed for a very long period of time to high glucose levels. However, glucotoxicity is considered to be a condition that results from chronic exposure (months or years) to high glucose levels, much higher than physiological values where it effectively generates damage to the organism in different tissues or organs. In our case the effects observed were only due to hyperglycemic stress that was maintained for a short or acute period of time (48 h) which would not imply the damage that has been reported for chronic pathologies such as glucotoxicity (Scheen M, Giraud R, Bendjelid K. Stress hyperglycemia, cardiac glucotoxicity, and critically ill patient outcomes current clinical and pathophysiological evidence. Physiol Rep. 2021 Jan;9(2):e14713. doi: 10.14814/phy2.14713.). Probably in these high glucose conditions a process such as autophagy is triggered (Bharath LP, Rockhold JD, Conway R. Selective Autophagy in Hyperglycemia-Induced Microvascular and Macrovascular Diseases. Cells. 2021 Aug 17;10(8):2114. doi: 10.3390/cells10082114.).

  1. In the manuscript the authors show that HG causes some necrosis but this is not affected by Feno, HR causes necrosis which is reduced by Feno and HR/GG causes very high levels of apoptosis which is eliminated by Feno. They show that PPARa is activated by Feno (as shown in ref 8), as were anti-apoptotic factors and that pro-apoptotic Bax was increased in HR, HG and HR/HG, which is interesting since Fig 6 showed that apoptosis was only detected in HG/HR. The authors then aim to show in Figures 10 and 11 that HG, HR and HR/HG caused mitochondrial damage but this is not particularly clear.

Answer

It should be considered that the treatment with FENO in the HG group was performed after the cells had been subjected to high glucose, so that the damage that may have been caused by the glucose can no longer be reversed by the drug treatment. In addition, we believe that our model does employ a high concentration of glucose but the time our cells are exposed to glucose is not for a prolonged period, so that the hyperglycemic stress that might be generated can be reversed at some point and that as mentioned in the previous paragraph may be activating processes involving Autophagy. This process tries to maintain cell homeostasis, regenerating damaged organelles or structures within the cell, such as mitochondria. Hence, this results in the increase of apoptotic proteins and the decrease of anti-apoptotic proteins when glucose levels are high. In fact, electron micrographs and MitoTracker fluorescence show, an altered structure and decreased mitochondrial activity indicating probably recycling of this organelle. This could be an interesting project for future studies, where we could study the mitochondrial dynamics, studying processes as biogenesis, fusion and fission mitochondrial, which would provide us a more accurate picture of what happens when we subject cardiac cells to high glucose in the absence of FENO (Bharath LP, Rockhold JD, Conway R. Selective Autophagy in Hyperglycemia-Induced Microvascular and Macrovascular Diseases. Cells. 2021 Aug 17;10(8):2114. doi: 10.3390/cells10082114.). In fact, these analyses are being carried out as a separate project in the laboratory which we hope to publish soon.

  1. In summary, this paper shows that fenofibrate activates PPARa can be protective against extreme ischaemia and high glucose in isolated cardiomyocytes. They claim this is due to anti-apoptotic factors but they only show that apoptosis is elevated in HR/HG. Furthermore, this has been shown in various animal models and so is not particularly novel.

Answer

This project is the work of a group of researchers who, like many groups put their effort and dedication to try to answer a question about a topic that has not been much studied, mostly if we talk about pharmacological preconditioning generated by FENO. This is a new topic with much potential for study in other cellular or animal models and that could be used in the clinical therapy in the future. The study in pathological conditions such as hypoxia and high glucose is also novel.

Points to clarify

  1. The authors show that HIF is present at the end of this protocol which is interesting since HIF protein is expected to be downregulated rapidly on re-oxygenation and suggests that oxygen is not delivered back to the cells through the cell culture media very efficiently

Answer

In fact, we evaluated the expression of Hypoxia Induced Factor (HIF 1a) as a marker of hypoxia, which was determined in our cardiac cells without reoxygenation. As you mention, reoxygenation causes a rapid attenuation of the expression of this protein.

  1. Figure 3 shows that HR, HG and HR/HG reduce cell viability (as shown previously in ref 8) but there are no indicators of significant differences from control cells in the cells without treatment

Answer

You are right, there is no difference in the control conditions, because the cells are not subjected to any stress (HG, HR or HGHR), although they are in the presence of FENO, DMSO or Mannitol (which is used as an osmotic control). In these conditions we do not expect any change in cell viability, which shows that none of these compounds have any consequence on the cells per se.

  1. Figure 4 shows the cells suffer nuclear and mitochondrial damage in HR/HG and HR/HG and this is rescued by Feno. At the start of paragraph 3.3 it says that the hallmark process of cytotoxicity is the secretion of granules but the methods say that cytotoxicity is measured via mitochondrial membrane potential with MITO-ID. The data in Fig 4A is presented as pmol/mg prot so it is not clear exactly what is shown.

Answer

MITO-ID is a Kit that serves to determine the transmembrane potential in the mitochondria through the accumulation of fluorescent particles that are displaced and accumulate in the form of aggregates or granules on one or the other side of the mitochondrial inner membrane, depending on the stability of the organelle. This is observed through the accumulation of these particles inside mitochondria; when the organelle is functioning properly, the indicator emits an orange fluorescence, but if the mitochondria has a low membrane potential or is damaged, the fluorescence emitted by the indicator is green. You are right, we made a mistake in the units of measurement in Figure 4, the correct way to express the changes in fluorescence is by Relative Fluorescence Units (This was indicated in the Methodology and in Figure 4, marked with yellow).

  1. Figure 10 shows labelling of mitochondria in ‘live and fixed cells’ – not sure what that means – and is suggested to show that this live mitochondria were reduced by HR, HG and HR/HG and increased using Feno but this is not clear from the images provided and should be quantified. Figure 11 (labelled as Figure 10) shows that treatment with HR, HG and HR/HG damages mitochondria and that Feno is protective but the images of HR-Feno and HG-Feno-treated mitochondria seem a very different size from controls or treated mitochondria.

Answer

What we mean by “live and fixed cells” is that the cells were incubated in the presence of MitoTracker when they were alive, as indicated by the technique, and after the incubation time, these cells were fixed with PFA for subsequent analysis under the microscope. We add in Figure 10, the graph of fluorescence quantification expressed in optical density (Figure 10).

We corrected the number of Figure 11, which we had mistakenly numbered as 10. Indeed a different morphology is observed in each treatment because the damage caused by HG and HR is very different, HR is a more aggressive condition than HG. In fact, it can be observed how the mitochondria with HG have less damage, are more defined and in most of the experiments a less damaging effect is observed in this condition, compared to HR, even so a very important protection is observed when FENO.

  1. Some aspects of the manuscript are over-detailed, I suspect they have been taken from a PhD thesis and could benefit from further editing. For example, in some methods (eg 2.10 and 2.11) unnecessary detail is provided.

Answer

The description of the methodology was not taken from a doctoral thesis, we considered a more detailed description of each technic used in this work. If you consider it necessary to resume this description, we will be happy to do so.

Reviewer 2 Report

Comments and Suggestions for Authors

interesting work that requires some clarifications 1- indicate the composition, including the amount of glucose, of the maintenance medium of the Neonatal Rat Cardiomyocytes cells used 2- the mannitol group treated with the different experimental protocols and analyses is missing. Since the state of the mitochondria is evaluated, the mannitol group must be present for reasons related to osmolarity 3- the mitochondrial purity markers are missing 4- insert an overall scheme that highlights how the intervention of FENO occurs from a molecular point of view

Author Response

Thank you very much for your comments and suggestions.

Comments and Suggestions for Author:

  1. Indicate the composition, including the amount of glucose, of the maintenance medium of the Neonatal Rat Cardiomyocytes cells used.

Answer

Neonatal cardiomyocytes after isolation were maintained in the following base medium described in reference 8 and before subdivision of the experimental groups: F-10 (1X) nutrient mixure (HAM) with L-Glutamine (Gibco Waltham, MA, USA) culture medium with 5.5 mM Glucose, supplemented with inactivated fetal bovine serum (Invitrogene, Carls-bad, CA, USA), 100 U/mL penicillin and 100 mg/L streptomycin (Gibco Waltham, MA, USA). We added the reference in section 2.2 of Methodology.

  1. The mannitol group treated with the different experimental protocols and analyses is missing. Since the state of the mitochondria is evaluated, the mannitol group must be present for reasons related to osmolarity

Answer

We considered that it was not necessary add Mannitol in all experiments because the concentration of Mannitol used in Control conditions gives similar results to those observed in the other conditions, as control (5.5 mM glucose), control-FENO and control-DMSO. For us, this was enough to prove this effect because it was demonstrated that the osmolarity of the medium in which our cells were maintained was within the osmolarity for the optimal range for the cells in culture. This means that there should not be change or effect inside the cell and therefore the mitochondria are not altered, this can also be seen when we measure Cyt c in the cytosol. If there was an effect by osmolarity in control conditions and without mannitol, the Cyt c levels in cytosol would be increased. Moreover, we would observe changes in the structure. On the contrary we did not detect Cit c in cytosol and  we observed well-defined mitochondria,  that were not swollen and the two membranes and cristae were well defined.

  1. The mitochondrial purity markers are missing

Answer

You are right in that observation; however, the objective of this work was to study only markers related to apoptosis and not mitochondrial homeostasis. According to the literature, the images by electron microscopy in control conditions can be taken as a good indicator of the purity of the isolated mitochondrial fraction, because in control conditions the two mitochondrial membranes and the internal structures such as the cristae are well defined and intact, conditions that change when the cells are subjected to high glucose or hypoxia-reoxygenation or both. Also, Cyt c in the cytosol or mitochondria can be considered as a good metabolic indicator of mitochondrial integrity. As seen in our results, a very weak signal of cytosolic Cyt c is observed in control conditions as opposed to mitochondrial Cyt c which is much higher, indicating that we have healthy and integrated mitochondria. This suggestion is a good point to study in the continuation of this work.

  1. Insert an overall scheme that highlights how the intervention of FENO occurs from a molecular point of view

Answer

We now insert a Graphical Abstract where we try to explain the function of fenofibrate in the pathological conditions. And we are also inserting it in the corresponding section.

Reviewer 3 Report

Comments and Suggestions for Authors

The manuscript focuses on the topical issue of myocardial pharmacological preconditioning. The study design is in line with the objectives of the study and the study itself is of a high methodological standard.
Minor comments
1 line 125 must be The treatment .......was 4h, instead of were 4 h.
2 line 272 must be .....comparisons we applied...., instead of ...comparison. We applied;
3 Is the same magification for b vs c in fig.10?

Author Response

Thank you very much for your comments and suggestions.

Comments and Suggestions for Author:

The manuscript focuses on the topical issue of myocardial pharmacological preconditioning. The study design is in line with the objectives of the study and the study itself is of a high methodological standard.

Minor comments

  1. Line 125 must be The treatment .......was 4h, instead of were 4 h.

Answer

We corrected this in the text.

  1. Line 272 must be .....comparisons we applied...., instead of ...comparison. We applied;

Answer

All ready corrected in the text.

  1. Is the same magnification for b vs c in fig.10?

Answer

Yes, all images were taken at the same size (500 nm, 50,000x). The difference observed in the images that you comment are due to the exposure to hypoxia-reoxygenation that affects most strongly the structure of subcellular particles as mitochondria because it is more aggressive than high glucose, as you can see in both images.

Round 2

Reviewer 1 Report

Comments and Suggestions for Authors

I apologise for the confusion over Feno and WY in my review and thankyou for clarifying your research and correcting a number of errors that made the manuscript confusing.

Figure 3 still needs correction as, at present, there are no significant differences between the control cells and the cells in HR, HG or HR/HG without treatment.

You still have two Figure 10s.

For the EM images the mitochondria you say:

This was observed as less damage in the membranes, decreased density, and preservation of normal size, indicating that the mitochondria are functional (Fig 11C, E, and G, respectively).

and in the discussion

As observed on Figure 11, the ultrastructure of mitochondria exhibits loss of mitochondrial cristae structure, expansion of the mitochondrial matrix, reduction in mitochondrial size, and hyperdensity of the mitochondrial matrix secondary to HR, HG, and HR/HG conditions.

The mitochondria in HR, HG and HR/G shown in Fig 11 were larger than control and didn't show a 'reduction in mitochondrial size'. In contrast the mitochondria in figures C and E are substantially smaller than the controls in A, which does not correspond to 'preservation of normal size'. Please comment on this in the results and suggest why Feno is leading to mitochondrial shrinkage, which you imply in the discussion is a bad thing.

Please enter text in the methods or results to indicate that the measurement of HIF was undertaken in cells that did not undergo 1 hour of reoxygenation

Author Response

Thank you very much for your comments and suggestions.

  1. I apologies for the confusion over Feno and WY in my review and thank you for clarifying your research and correcting a number of errors that made the manuscript confusing.

Answer

Thank you very much. Your comments have been incredibly helpful in enriching our work and have provided us with new ideas for future projects.

  1. Figure 3 still needs correction as, at present, there are no significant differences between the control cells and the cells in HR, HG or HR/HG without treatment.

Answer

We corrected the image and the description in the footer.

  1. You still have two Figure 10s.

Answer

You were right; we forgot to change the number in the figure, but it has now been corrected.

  1. For the EM images the mitochondria you say:

This was observed as less damage in the membranes, decreased density, and preservation of normal size, indicating that the mitochondria are functional (Fig 11C, E, and G, respectively).

and in the discussion

As observed on Figure 11, the ultrastructure of mitochondria exhibits loss of mitochondrial cristae structure, expansion of the mitochondrial matrix, reduction in mitochondrial size, and hyperdensity of the mitochondrial matrix secondary to HR, HG, and HR/HG conditions.

The mitochondria in HR, HG and HR/G shown in Fig 11 were larger than control and didn't show a 'reduction in mitochondrial size'. In contrast the mitochondria in figures C and E are substantially smaller than the controls in A, which does not correspond to 'preservation of normal size'. Please comment on this in the results and suggest why Feno is leading to mitochondrial shrinkage, which you imply in the discussion is a bad thing.

Answer

You are correct; we made a mistake regarding the effect of Feno in both the description and discussion of its impact on mitochondria. In the results section, we intended to convey that the presence of Feno prevents mitochondrial damage even under pathological conditions such as HG, HR, and both, thereby conserving mitochondrial morphology.

In the discussion, we should NOT have included the phrase “reduction in mitochondrial size.” As shown in the images, when cells are subjected to HG, HR, and both conditions, mitochondria actually swell, and both the inner and outer membranes lose their original morphology, indicating mitochondrial damage in these conditions. We correct this idea in the text.

The presence of feno promotes a reduction in oxidative stress (Reference 8), inhibiting apoptosis and leading to shrinkage in mitochondria due to decreased ion exchange across the mitochondrial inner membrane. This ultimately protects both mitochondrial morphology and function.

  1. Please enter text in the methods or results to indicate that the measurement of HIF was undertaken in cells that did not undergo 1 hour of reoxygenation

Answer

We did the mention about HIF in results.

Reviewer 2 Report

Comments and Suggestions for Authors

I agree new version

Author Response

Thank you very much for your time in reviewing this paper.